# Survival and Treatment Outcomes in Gastric Cancer Patients with Brain Metastases: A Systematic Review and Meta-Analysis [note 1]

**DOI:** 10.3390/cancers16223796

**Published:** 2024-11-12

**Authors:** Daniel Sur, Adina Turcu-Stiolică, Emil Moraru, Cristian Virgil Lungulescu, Cristina Lungulescu, Vlad Iovanescu, Petrica Popa

**Affiliations:** 1Department of Medical Oncology, The Oncology Institute “Prof. Dr. Ion Chiricuţă”, 400015 Cluj-Napoca, Romania; daniel.sur@umfcluj.ro; 2Department of Medical Oncology, University of Medicine and Pharmacy “Iuliu Hațieganu”, 400012 Cluj-Napoca, Romania; 3Department of Pharmacoeconomics, Faculty of Pharmacy, University of Medicine and Pharmacy of Craiova, 200349 Craiova, Romania; adina.turcu@umfcv.ro; 4Surgery Department, University of Medicine and Pharmacy Craiova, 200349 Craiova, Romania; emil.moraru@umfcv.ro; 5Department of Medical Oncology, University of Medicine and Pharmacy Craiova, 200349 Craiova, Romania; 6Oncolab, 200385 Craiova, Romania; cristina.lungulescu@yahoo.com; 7Department of Gastroenterology, Research Center of Gastroenterology and Hepatology, University of Medicine and Pharmacy of Craiova, 200349 Craiova, Romania; vlad.iovanescu@umfcv.ro (V.I.); petrica.popa@umfcv.ro (P.P.)

**Keywords:** gastric cancer, brain metastases, surgery, radiotherapy, HER2, overall survival, incidence

## Abstract

Brain metastases from gastric cancer are rare but pose serious challenges for patients and doctors due to their impact on survival and quality of life. Despite improvements in cancer treatment, brain metastases remain difficult to treat, and there is little clear guidance on the best approaches. This study aims to gather and analyze the available research to better understand how common brain metastases are in gastric cancer patients, what factors may increase the risk, and how different treatments, including surgery, affect patient survival. By providing clearer information, this research hopes to guide doctors in choosing the most effective treatments and highlight areas where further study is needed.

## 1. Introduction

Gastric cancer ranks as the fifth most commonly diagnosed malignancy and is the fourth leading cause of cancer-related mortality worldwide, based on the most recent information provided by GLOBOCAN [1]. Gastric tumors are highly heterogenous and occur through a variety of genetic abnormalities and environmental predisposing factors, including Helicobacter pylori infection, eating patterns, and smoking [2]. As a result of its propensity for metastasis and early-stage non-specific symptoms that frequently lead to underdiagnosis, gastric cancer (GC) poses numerous clinical challenges [3,4]. Metastasis to the liver is a common occurrence in gastric malignancies, due to the liver’s extensive blood supply and its role as a first-pass organ for cells disseminating through the portal circulation. Other common sites include the peritoneum, lymph nodes, lungs, and bones [5]. Traditionally rare, with an incidence of less than 1% [6,7,8], brain metastases (BM) are frequently described as a late manifestation of gastric cancer that severely affect patient prognosis and management [9].

Following the initial documented case in 1960 [10], the literature has portrayed gastric cancer patients with cerebral involvement as a highly diverse population due to the challenging natural evolution of the disease [11].

Patients with gastric cancer who develop brain metastases typically experience a shorter median overall survival (mOS), averaging around 5.3 months (ranging from 2 to 9.6 months), compared to patients with brain metastases originating from other cancers, like those of the lung, breast, or kidney [12]. Additionally, the prognosis in gastric cancer cases with brain metastases is affected by the presence of metastases in other locations, such as the liver and lungs, as well as by the advancement of systemic disease [13].

Tumor metastasis progresses through four main stages: invasion, entry into the bloodstream, passage through the blood–brain barrier (BBB), and, finally, colonization. The epithelial–mesenchymal transition (EMT) involved in this process is influenced by various regulatory factors, including RAS and Hedgehog [14,15]. For tumor cells to metastasize to the brain, they must penetrate tight junctions and other structural barriers. During this process, microglia activate astrocytes to increase TNF levels through the effects of TNF-α, IL-1β, IL-1α, and other pro-inflammatory mediators [16,17].

Brain metastases often develop after cancer has progressed significantly and spread to other common sites, although they may be synchronous or even manifest as an initial symptom [18]. In addition, BM might occur after the successful treatment of gastric cancer, either early or after a considerable period of time [19,20]. Recent data indicate that the frequency of BM may be underestimated [21], a phenomenon likely attributable to improved survival rates due to advancements in systemic therapies and more sensitive diagnostic tools [20].

The presence of brain metastases requires complex therapeutic approaches, and specific management is highly dependent on various factors including the patient’s performance status, the number and site of metastases, and dissemination to other organs. Key treatment options include neurosurgery, stereotactic radiosurgery, palliative radiotherapy, and chemotherapy [22]. Most systemic chemotherapeutics have poor blood–brain barrier penetration, limiting the treatment of GC brain metastases [23]. BM often lead to the early onset of neurological symptoms such as headaches, cognitive impairments, and motor deficits. The presence of BM drastically deteriorates the quality of life (QoL) of patients, due to both the physical and psychological burden of the disease [24]. The management of symptoms and the preservation of QoL become integral parts of the treatment strategy, alongside the direct treatment of the metastases themselves.

The current literature on BM from gastric cancer is fragmented and inconclusive, with existing studies that often present conflicting findings regarding the incidence, risk factors, clinical presentation, and outcomes of BM in gastric cancer, creating a need for a systematic collation and analysis of available data. This systematic review and meta-analysis aims to consolidate existing research to provide a clearer understanding of the incidence and clinical implications of BM in gastric cancer. Specifically, we seek to quantify the incidence of BM in gastric cancer patients, identify common clinical presentations and risk factors associated with BM, evaluate the impact of BM on patient outcomes, and assess the effectiveness of current management strategies, including surgical and non-surgical interventions. This meta-analysis aspires to contribute to the understanding of BM in GC, enhancing clinical management and guiding future research to improve patient care and outcomes in this challenging field.

## 2. Materials and Methods

We performed this systematic review according to the Preferred Reporting Items for Systematic Reviews and Meta-Analyses (PRISMA) guidelines [25], as we present the PRISMA 2020 Checklist in Appendix A. The protocol of our systematic review was registered in the PROSPERO platform [26] with the number CRD42021256696.

### 2.1. Criteria Considered in This Review

We searched online databases for observational or randomized controlled studies on gastric cancer patients with brain metastases. The inclusion and exclusion criteria for the studies were clearly defined to ensure the selection of high-quality and relevant research. Studies that were retrospective or prospective in design and involved patients with a diagnosis of both gastric cancer (GC) and brain metastases (BM) were eligible for inclusion. Additionally, we included studies that evaluated key interventions such as surgical resection, radiotherapy (including stereotactic radiosurgery or whole-brain radiotherapy), or chemotherapy in the treatment of BM. Studies reporting on symptom presentation, the correlation between HER2 status and BM, and clinical outcomes including overall survival (OS) and progression-free survival (PFS) were also included.

In contrast, studies that did not meet these criteria were excluded. The exclusion criteria applied during the review process encompassed duplicate records, studies with different outcomes of interest or insufficient data, and studies published in languages other than English. Research that focused on cancers other than gastric cancer or studies that included mixed primary cancer types without separating GC data were also excluded.

### 2.2. Research Identification Retrieval Methods

All published studies in English were identified by conducting a literature search in the electronic databases MEDLINE (1966 to date of search), Web of Science, and Scopus from inception to August 2023, using the following search: ((gastric) AND ((cancer) OR (tumor) OR (malignancy))) AND ((brain) AND (metastases)). We checked the reference lists of the included research and previous systematic reviews to determine further research reports.

We imported titles and abstracts retrieved by the searches into Excel. Duplicate records were identified and removed. The titles and abstracts of all potential studies were screened by two review authors (D.S. and V.I.), and inconsistencies were discussed with the third reviewer (P.P.) until consensus. Next, two review authors (C.V.L and C.L) independently screened full-text articles for inclusion. Again, in the case of disagreement, consensus was reached on inclusion or exclusion by discussion, and if necessary, the third researcher (A.T-S.) was consulted. The data collection process followed the same procedure, extracted data being compared, with any discrepancies being resolved through discussion. We extracted information relating to the characteristics of the included studies and results as follows:Study identifiers and the characteristics of the study design (study name, location, study design).The characteristics of the groups (number of patients with GC and BM, sex, presenting symptoms).Interventions: chemotherapy, radiotherapy, surgery.Comparison: overall survival (OS), progression-free survival (PFS).

### 2.3. Statistical Analysis

The characteristics of the patients were analyzed with the mean (95% confidence intervals, CIs) or median for continuous data and percentage for discrete data. A narrative summary was provided if it was not appropriate for a meta-analysis to be performed.

The meta-analysis employed the standardized mean difference as the primary outcome measure, and a random-effects model was applied to the data in the case of high heterogeneity. The level of heterogeneity, expressed through tau^2^, was estimated using the restricted maximum likelihood method. Alongside the tau^2^ estimate, both the Q-test for heterogeneity and the I^2^ statistic were reported. Whenever heterogeneity is detected (as indicated by the tau^2^ estimate, regardless of the Q-test outcome), a prediction interval for the true effect sizes is provided. To assess whether any studies might be outliers or exert undue influence on the model, Studentized residuals and Cook’s distances were examined. Studies with Studentized residuals exceeding the 100 × (1 − 0.05/(2 × k))th percentile of a standard normal distribution were flagged as potential outliers, using a Bonferroni correction (with a two-sided alpha of 0.05 for k studies). Additionally, studies with a Cook’s distance greater than the median plus six times the interquartile range were deemed influential. The rank correlation Begg’s test and Egger’s regression test were used to check for funnel plot asymmetry by checking for significant publication bias. Statistical analyses were performed by Review Manager (Rev Man, version 7.2.0, The Cochrane Collaboration, London, UK, 2024) and R package Metafor (version 4.1, R foundation, Vienna, Austria). A *p*-value < 0.05 was considered statistically significant.

The Newcastle–Ottawa Quality Assessment Scale (NOS) was used to assess the quality of non-randomized studies, which allocates a maximum of nine points for the quality of selection, comparability, exposure, and outcome of study participants [27]. Studies with NOS scores 0–3, 4–6, and 7–9 were considered as low-, moderate-, and high-quality, respectively [28].

## 3. Results

The database search identified no previous version of a review with the same objectives as ours. A total of 1740 publications were identified through the initial literature search, and 1317 studies remained after duplications were excluded. Following the title and abstract screens, 1266 publications were removed because they were not related to the specific objective. Fifty-one potentially relevant articles were identified for the comprehensive review. Following the full-text assessment, 17 studies [7,9,29,30,31,32,33,34,35,36,37,38,39,40,41,42] were included in the final systematic review and meta-analysis, as shown in Figure 1.

Out of the 17 full-text papers, 16 were retrospective studies, and 1 was a prospective study [39]. All the studies scored at least 7 for the NOS, which proves they were high-quality, according to the internationally accepted ranking. The characteristics for each study included are integrated in Table 1.

### 3.1. Incidence

We identified 12 studies that reported the incidence of BM due to GC [29,30,31,33,43]. In total, 70,237 GC patients were included, with 621 of them diagnosed with BM. The average incidence of BM in GC patients was 2.29% (95%CI from 1.06% to 3.53%), ranging from 0.47% to 7.79%, as depicted in Figure 2.

Only five studies focused on subgroup analysis by gender. A total of 69.4% were males, and 30.6% were females, as shown in Table 1. Zhang et al. performed a subgroup analysis by race and categorized patients as White (82.3%), Black (8.2%), or other races (9.5%) [29]. Qiu et al. only stated that Caucasian patients (84.1%) had a higher percentage (*p*-value = 0.002) of BM than African American (6.6%) or Asian patients (9.3%) [33]. York et al. retrospectively reviewed GC patients with BM, among which 79.2% were Caucasian, 4.2% were Black, and 16.7% were Hispanic [7].

### 3.2. Symptoms

Only seven studies presented the neurological symptoms of CNC involvement, such as headache, gait disturbance, muscular weakness, hemiparesis, aphasia, and visual troubles [7,9,32,35,39,41,43]. Some of them showed that all BM patients with GC had neurological symptoms [32]. Brain metastases were discovered after the appearance of neurological symptoms [35], but sometimes, BM were asymptomatic [39]. Headache was a common symptom experienced by patients with GC and BM (five of the seven studies presented it with a mean of 37%, ranging from 18% to 57%), as shown in Table 2.

### 3.3. HER2 Status

HER2-positive status appears to be associated with a higher risk of developing BM. The HER2 status of GC was assessed in only two studies [32,35]. However, Cavanna et al. only provided the HER2 status information for patients with CNS metastases, which could not be included in the meta-analysis [32]. The odds ratio was 43.24 (95%CI 2.05–913.39), indicating a statistically significant difference between patients with a positive HER2 status and those with a negative HER2 status (*p* = 0.02), as shown in Figure 3.

### 3.4. Survival Time

The mean time from the diagnosis of gastric or GE junction cancer to the diagnosis of CNS metastases in the 67 patients among 2671 GC patients was 11.44 months, as shown in Table 3. The mean OS time from the diagnosis of GC in the 18 patients with CNS metastases was 14.2 months, while the median OS time from the diagnosis of CNS metastases was 4.72 months. The 1-year and 5-year OS was 17.6% [9,29] and 0%, respectively [29,33]. Only one study compared the efficacy of gamma knife radiosurgery (GKR) versus WBRT, resulting in a more favorable median survival after BM for GKR (40 weeks, 95% CI from 44.9 to 132.1 weeks) than that for WBRT (9 weeks, 85% CI from 8.8 to 21.9 weeks) [40].

### 3.5. Meta-Analysis of Overall Survival Time of Patients Who Underwent Surgical Resection for BM Versus Those Without Surgical Resection

A total of four studies [7,39,41,43] were included in this analysis, with a total of 9 patients with BM and GC treated with resection and 36 patients treated with radiosurgery or WBRT. The observed mean differences ranged from 3.8 to 27.5, with the majority of the estimated mean difference (MD) being positive (100%). According to the Q-test, there was no significant amount of heterogeneity in the true outcomes (tau^2^ = 78.77, Q(3) = 19.71, *p* = 0.0002, I^2^ = 85%), as shown in Figure 4. Based on the random-effects model, the pooled MD was 12.39 (95%CI: 2.03–22.75), with a statistically better OS for patients who underwent surgical resection than those without surgical resection (*p*-value = 0.02).

An examination of the Studentized residuals revealed that none of the studies had a value larger than ±2.4977, and hence, there was no indication of outliers in the context of this model. According to the Cook’s distances, none of the studies could be considered to be overly influential. Neither the Begg rank correlation (*p* = 0.7500) nor the Egger’s regression test (*p* = 0.8299) indicated any funnel plot asymmetry, as shown in Figure 5.

## 4. Discussion

Gastric cancer is associated with high morbidity and mortality rates, underlining the necessity for new therapeutic interventions based on previous clinical knowledge [44]. GC is often diagnosed at advanced stages, contributing to its poor prognosis and high mortality rate. Due to its heterogeneous presentation and the often late onset of symptoms, there is a critical need for effective biomarkers that can improve early detection and guide treatment decisions. While classical biomarkers like CEA and CA 19-9 have long been used, novel markers such as miR-106 are being explored to enhance diagnostic accuracy and potentially reveal distinct tumor biology. Although miR-106 shows potential, established markers like CA 19-9 continue to be reliable predictors and may significantly bolster early detection efforts [45]. Accurate early diagnosis is crucial in managing gastric cancer and its metastatic complications, particularly for conditions like brain metastases that often remain undetected until symptoms appear. Just as endoscopic ultrasound is an essential tool for diagnosing rare conditions [46], this imaging method plays a similarly valuable role in facilitating early detection and supporting personalized treatment planning in gastric cancer [47]. Although brain metastases from gastric cancer are only present in <1% of patients with metastatic gastric cancer, these rare and late manifestations do not have a standard therapeutic scheme. The prognosis is poor with a median overall survival of approximately 3 months. Current treatment algorithms are based on small studies that individualize therapy considering the personal characteristics of the patient [6,9].

The current systematic review and meta-analysis is the first to analyze the clinical outcome of gastric cancer patients with brain metastases using data from retrospective and prospective studies. Most of the studies presented were retrospective studies. In terms of incidence, our analysis found that BM occurred at an average incidence of 2.29% (95% CI: 1.06% to 3.53%), ranging from 0.47% to 7.79. These findings match those of Lin et al. [48] who reported a 3% incidence based on an analysis of the SEER database. In addition, Namikawa et al. [49] reported an incidence of 2,9%, which is consistent with general reports. The explanation for the marginal differences in the percentages could be because the GC population was highly selective, and a good percentage of patients had more than one distant localization of metastases. This could signify very advanced disease and increase their risk of developing brain metastases [5].

Scarce data were found in the literature on the symptoms of brain metastases from gastric cancer. Headache, weakness, changed mental status, focal neurological impairments, gait or visual problems, and ataxia were typical signs of brain metastases. CNS metastases can occasionally go unnoticed [7,9]. The findings of our study support the data from previous studies considering that symptoms due to brain metastases can appear during the evolution of the disease, but this is not mandatory given that some patients can be asymptomatic.

The role of HER2-positive status in BM from gastric cancer is increasingly recognized. Both Cavanna et al. [32] and Blay et al. [35] emphasize the higher incidence of BM in HER2-positive patients compared to HER2-negative ones. Cavanna et al. observed that 85.71% of patients with CNS metastases from GC had HER2-positive tumors, suggesting a strong association between HER2 overexpression and CNS involvement [32]. Blay et al. further confirmed this with an odds ratio of 43.24 (95% CI 2.05–913.39, *p* = 0.02), highlighting a statistically significant difference in BM occurrence between HER2-positive and HER2-negative patients. Furthermore, out of the 11 HER2-positive patients, 3 (27%) developed brain metastases, while none of the 52 HER2-negative patients did (*p* = 0.004). The brain metastases were identified following the onset of neurological symptoms such as motor deficits, headaches, and strokes [35]. Additionally, none of the HER2-positive patients developed peritoneal carcinomatosis, unlike 39% of the HER2-negative group (*p* = 0.013). There was no notable association between HER2 status and other metastasis sites [35].

In accordance with previous evidence that those with HER2-positive gastric cancer tumors have a higher percentage of developing brain metastases, our analysis found that HER2 status appears to be associated with a higher risk of having cerebral metastases [35,50]. Tinckell et al. showed that the risk of developing brain metastasis was significantly increased in patients with GC and HER2-positive status [51]. These findings mandate a more thorough evaluation of symptoms for patients with HER2-positive GC [51].

In both studies assessing HER2 status, the overall survival (OS) outcomes for HER2-positive gastric cancer patients with brain metastases were poor, despite multimodal treatment approaches [32,35]. Blay et al. [35] stated that treatment for all three HER2+ patients with BM involved a combination approach—surgery for one, radiotherapy for all, and chemotherapy with trastuzumab. The response to treatment was varied, with one patient experiencing partial remission and the other two showing stable disease. An exploratory analysis of progression-free survival (PFS) showed that HER2-positive patients treated with trastuzumab had a significantly longer PFS (9.4 months) compared to HER2-negative patients (5.0 months, *p* = 0.041). However, while HER2-positive patients had a longer overall survival (OS), the difference was not statistically significant (*p* = 0.057) [35].

In the study by Cavanna et al., six out of seven patients (85.71%) with central nervous system (CNS) metastases from gastric cancer were found to be HER2-positive based on immunohistochemistry. Patients were treated with whole-brain radiation therapy (WBRT), and four of them also received systemic chemotherapy. Among these, two patients were treated with a combination of trastuzumab and chemotherapy, with one case involving intrathecal chemotherapy alongside trastuzumab for leptomeningeal carcinomatosis. One patient received only palliative care. The median time from the initial diagnosis of gastric or gastroesophageal junction cancer to the development of CNS metastases was 6.2 months. The overall median survival from the diagnosis of gastric cancer was 9.4 months, and the median survival from the detection of CNS metastases was 4.1 months [32]. Another study showed that GC tumors with HER2-positive expression can benefit from anti-HER2 therapies that can prolong the time to brain involvement [52]. These results highlight the challenges in treating BM in HER2-positive patients despite the use of trastuzumab, emphasizing the need for further research into more effective CNS-directed therapies. Furthermore, data on HER2-positive status and its link to BM are limited, as only two studies provided relevant information, restricting a more thorough statistical analysis.

However, both the Cavanna et al. and Blay et al. studies on HER2-positive gastric cancer patients with brain metastases have notable limitations regarding treatment approaches, primarily due to the treatment standards at the time. In the Cavanna study, patients were treated predominantly with whole-brain radiation therapy and systemic chemotherapy, with trastuzumab used in only two cases. This treatment protocol does not fully align with the current standard of care, which, since the ToGA trial in 2010, has recommended first-line trastuzumab in combination with platinum-based chemotherapy for HER2-positive gastric cancer [53]. Similarly, Bang et al. [53] reported multimodal treatment involving trastuzumab and surgery, but the timing and application of trastuzumab were inconsistent. Additionally, neither study included more recent therapeutic advancements such as trastuzumab deruxtecan (T-DXd), which has shown efficacy in HER2-positive gastric cancer after trastuzumab resistance [54,55].

In our analysis, the median OS from the moment of CNS metastasis diagnosis was 4.72 months. The 1-year OS was 17.6% [9,29], and the 5-year OS was 0 [29,33]. According to relevant studies, most patients had systemic metastases to other organs before brain metastases emerged. Moreover, numerous brain metastases were discovered in approximately half of these patients. The overall survival rate reported varied from 1.4 to 27.7 months. It is interesting to note that every study discovered that patients could have a higher chance of survival with aggressive treatment which included the surgical removal of brain metastases [6]. Because some studies have shown that patients with metastatic GC who underwent surgery had a relatively poor prognosis [56], we extracted the OS data for patients with and without surgery, without having additional information about the stage of GC [57]. Patients who underwent surgical resection had a longer OS from the diagnosis of BM (DM = 12.39, 95% CI: 2.03–22.75) compared with patients without resection. However, comparisons between treatment modalities, such as surgery and radiotherapy, remain challenging due to variability in protocols and patient selection, with some studies using whole-brain radiation therapy (WBRT) and others using stereotactic radiosurgery or surgery.

From a SEER database study, 5.1% of patients had bone metastasis, whereas 0.8% had brain metastasis. Patients with and without bone metastasis were found to have cause-specific survival rates of 1.3% vs. 29.9% (median of 4 months for bone metastasis), and 2.3% vs. 28.7% (median of 3 months for brain metastasis). BM in gastric cancer are frequently accompanied by extracranial metastases rather than occurring in isolation, as the majority of patients with BM also presented with metastases in other locations, such as the liver or lungs [33]. However, a small subset of studies documented cases of brain metastases appearing as an early manifestation or synchronously with primary gastric cancer [18,19]. These studies provide examples where BM do not strictly follow the pattern of late-stage progression, highlighting the variability in metastatic timing that can occur in gastric cancer patients.

Findings from this review indicate that patients with gastric cancer and brain metastases have a smaller incidence compared with patients with other metastatic lesions. Also, the common neurological symptoms are not necessarily indicative of brain metastases. Mostly, headache occurs in the majority of patients before the diagnosis of cerebral involvement. The HER2-positive status in gastric patients appears to correlate with a greater probability of developing brain metastases. Patients with gastric cancer who undergo metastasectomy for brain metastases have a higher overall survival than patients that do not receive surgery.

Recent therapeutic advancements for HER2-positive cancers are showcasing promising cross-applicability from breast to gastric cancers, emphasizing the potential of CNS-penetrating drugs. Trastuzumab deruxtecan, with significant intracranial efficacy, is now a second-line treatment for HER2-positive gastric cancer as per NCCN Clinical Practice Guidelines in Oncology (NCCN Guidelines^®^) [54,58]. Complementing this, tucatinib’s potential is being explored in the MOUNTAINEER-02 trial for its effectiveness in treating gastric cancer [59]. Furthermore, margetuximab, recently approved for use in HER2-positive breast cancer, is under assessment for gastric cancer through the MAHOGANY cohort A trial [60]. Together, these developments reflect a strategic initiative to harness proven breast cancer therapies for broader oncological applications, particularly focusing on treatments that address complex challenges like CNS metastases, potentially transforming the therapeutic landscape for gastric cancer.

This study has several limitations that should be acknowledged. First, there is significant heterogeneity among the included studies, particularly regarding patient populations, treatment modalities, and outcome reporting, which may impact the applicability to a broader population. Additionally, due to a lower-than-anticipated number of studies providing data on HER2-positive status, we could not perform a more powerful meta-analysis, limiting the strength of the conclusions drawn in this area. This study also lacks data on quality-of-life outcomes for patients undergoing different treatments, which is a critical factor in clinical decision-making for brain metastases. Finally, this review predominantly focuses on traditional treatments such as surgery and radiotherapy, with a limited discussion of novel therapies like targeted treatments or immunotherapies [55], which may play an increasing role in the future management of brain metastases from gastric cancer. Despite the current limitations, these results consisting of the analysis of a high number of patients with gastric cancer and the quality of the studies evaluated provide solid evidence on the clinical outcome and underline the need for additional research in the field.

## 5. Conclusions

In this work, the study presented in [61] is expanded upon. This systematic review and meta-analysis highlights that gastric cancer patients with brain metastases who undergo surgical excision experience improved overall survival compared to those who do not receive surgery for BM. Additionally, HER2-positive gastric cancer patients demonstrate a higher incidence of brain metastases. Further clinical and translational research is warranted to elucidate the molecular mechanisms in the patient subset who develop brain metastases, aiming to advance personalized treatment strategies.

## Figures and Tables

**Figure 1 cancers-16-03796-f001:**
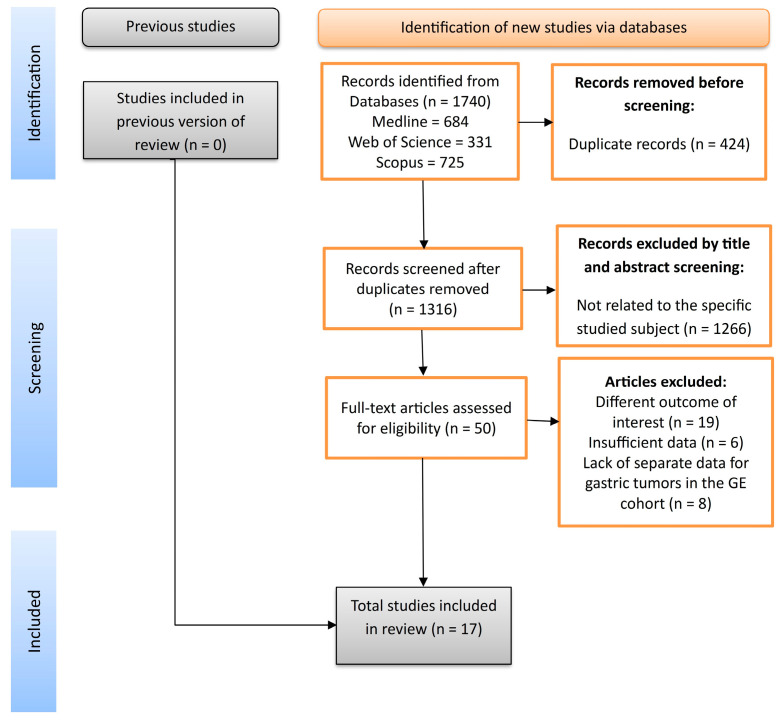
PRISMA flowchart.

**Figure 2 cancers-16-03796-f002:**
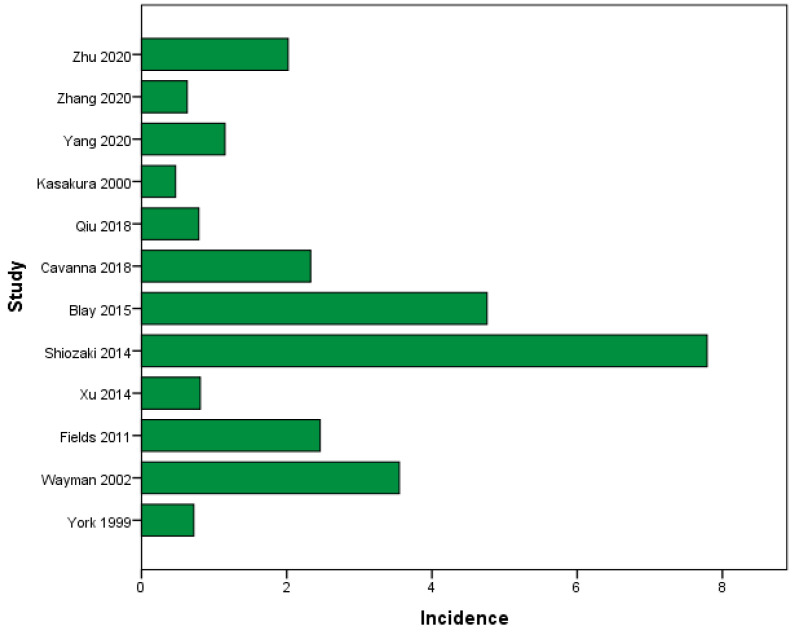
Incidence of BM (%) in all GC patients [7,29,30,31,32,33,35,36,38,39,42,43].

**Figure 3 cancers-16-03796-f003:**
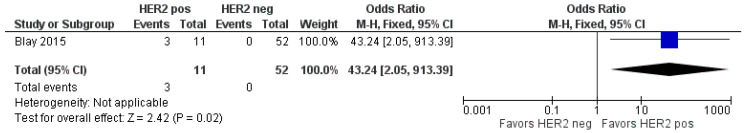
Forest plot comparison of brain metastasis patients with positive HER2 status and negative HER2 status [35].

**Figure 4 cancers-16-03796-f004:**
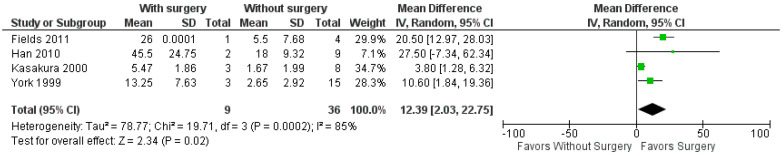
Forrest plot for comparing OS from diagnosis of BM in patients who underwent surgical resection with those without surgical resection [7,39,41,43].

**Figure 5 cancers-16-03796-f005:**
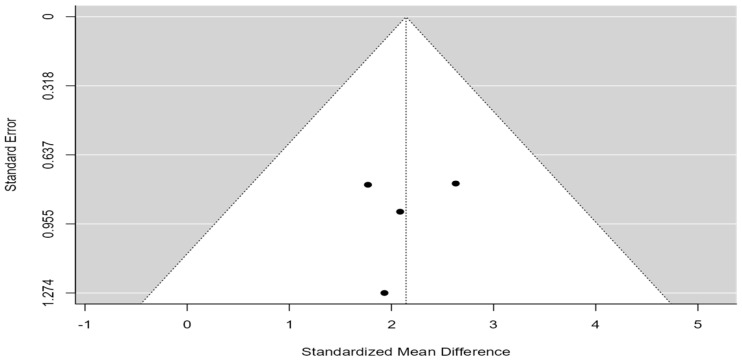
Funnel plot comparing OS from diagnosis of BM in patients who underwent surgical resection for BM with those without surgical resection.

**Table 1 cancers-16-03796-t001:** The characteristics of the included studies.

	Study with Reference	Country	Type of Study	Number of GC Patients	Number of BM	Male (n, %)	Female (n, %)	Quality Assessment (NOS)
1	Zhang et al. (2020) [29]	US	Retrospective	36.588	231	168 (72.7%)	63 (27.3%)	8
2	Zhu et al. (2020) [30]	US, China	Retrospective	4.795	97	NR	NR	9
3	Yang et al. (2020) [31]	US	Retrospective	2.174	25	NR	NR	9
4	Cavanna et al. (2018) [32]	Italy	Retrospective	300	7	3 (42.9%)	4 (57.1%)	8
5	Qiu et al. (2018) [33]	US	Retrospective	19.022	151	113 (74.8%)	38 (25.2%)	8
6	Rades et al. (2017) [34]	Germany	Retrospective	22	22	19 (86.4%)	3 (13.6%)	7
7	Kraszkiewicz et al. (2015) [9]	Poland	Retrospective	16	16	11 (68.8%)	5 (31.2%)	7
8	Blay et al. (2015) [35]	France	Retrospective	63	3	NR	NR	8
9	Shiozaki et al. (2014) [36]	USA	Retrospective	629	49	NR	NR	8
10	Minn et al. (2014) [37]	Republic of Korea	Retrospective	8	8	5 (62.5%)	3 (37.5%)	7
11	Xu et al. (2014) [38]	China	Retrospective	246	2	NR	NR	8
12	Fields et al. (2011) [39]	USA	Retrospective	609	10	NR	NR	9
13	Park et al. (2011) [40]	Republic of Korea	Retrospective	56	56	44 (78.6%)	12 (21.4%)	9
14	Han et al. (2010) [41]	Republic of Korea	Retrospective	11	11	7 (63.6%)	4 (36.4%)	7
15	Wayman et al. (2002) [42]	UK	Prospective	169	6	NR	NR	8
16	Kasakura et al. (2000) [43]	Japan	Retrospective	2.322	11	9 (81.8%)	2 (18.2%)	7
17	York et al. (1999) [7]	US	Retrospective	3.320	24	18 (75%)	6 (25%)	8

NR, not reported.

**Table 2 cancers-16-03796-t002:** Symptoms of patients with BM from GC.

Study	York et al. (1999) [7]	Cavanna et al. (2018) [32]	Kasakura et al. (2000) [43]	Blay et al. (2015) [35]	Kraszkiewicz et al. (2015) [9]	Fields et al. (2011) [39]	Han et al. (2010) [41]
Weakness	67%						
Headache	42%	57%	NR	NR	31.25%		18%
Nausea	38%		NR				
Seizures	8%					73%	
Gait disturbance	42%	57%					
Hemiparesis		29%		NR		45%	
Aphasia		29%					
Muscular weakness		57%			37.5%		
Visual trouble		14%	NR			27%	
Epilepsy			NR		12.5%		
Loss of consciousness					18.75%		

NR, not reported.

**Table 3 cancers-16-03796-t003:** Study characteristics on treatment and OS.

Study	Treatment			Median Follow-Up (Months)	Median Time from Diagnosis of GC to Brain Metastasis (Months)	Median OS from Diagnosis of GC (Months)	1-Year OS	5-Year OS	Median OS from Diagnosis of BM (Months)
	Surgery	Radiation	Chemotherapy						
Zhang et al. (2020) [29]	x	x	x				16.2%	0%	
Zhu et al. (2020) [30]	x	x	x						
Yang et al. (2020) [31]	x	x	x	3					
Cavanna et al. (2018) [32]		x			6.2	9.4			4.1
Qiu et al. (2018) [33]	x							0%	3
Rades et al. (2017) [34]	x	x		9.4	10				
Blay et al. (2015) [35]	x	x	x	55					
Kraszkiewicz et al. (2015) [9]	x	x	x	52	12.3		19%		2.8
Minn et al. (2014) [37]	x								
Xu et al. (2014) [38]		x	x						
Shiozaki et al. (2014) [36]	x	x	x	37.2					
Fields et al. (2011) [39]	x	x	x	46					
Park et al. (2011) [40]	x	x	x		GKR = 6/WBRT = 11				GKR = 10/WBRT = 2.1
Han et al. (2010) [41]	x	x	x		23	19			15
Wayman et al. (2002) [42]	x								
Kasakura et al. (2000) [43]	x	x	x		5.7				1.3
York et al. (1999) [7]	x	x	x		4				2.5

OS, overall survival; TMT, trimodality = chemoradiation then surgery; BMT, bimodality = chemoradiation; x, treatment type.

## Data Availability

The data that support the findings of this study are openly available.

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
