# Peer review of "Survival and Treatment Outcomes in Gastric Cancer Patients with Brain Metastases: A Systematic Review and Meta-Analysisâ€"

_cancers, 2024, doi:10.3390/cancers16223796_

Round 1
Reviewer 1 Report
Comments and Suggestions for Authors
The review "Survival and Treatment Outcomes in Gastric Cancer Patients with Brain Metastases: A Systematic Review and Meta-analysis" discusses patient outcomes, with a focus on brain metastases, a relatively underexplored secondary determination in the literature. Here are my recommendations:
1. The introduction is too brief and lacks sufficient literature background. It should be expanded.
2. The Materials and Methods section should include a flow chart, currently found in the Results section.
3. Lines 122-129 should be removed.
4. In the tables, cite the reference only, without including the author's name.
5. Discuss ongoing clinical trials related to this topic in more detail.
6. Why did you choose a systematic review and specifically, why conduct a meta-analysis on this topic? The results are quite weak, especially due to the small number of studies. This undermines the reliability of the results and the strength of the study.
7. Table 3 should be removed.
8. The Discussion section should address unclear information from the literature and critically analyze it, including limitations of the studies in the literature. I recommend also discussing new diagnostic methods for gastric cancer using microRNA (10.3390/ijms25147898) and new differential diagnostic methods for gastric pathology via endoscopic ultrasound, including rare pathologies (10.3390/diagnostics14070675).
9. The conclusions should be reformulated and the first sentence removed.
10. It would be appropriate to add a paragraph on future perspectives.
Reviewer 2 Report
Comments and Suggestions for Authors
Interesting paper and review, excellent work.
1. Do brain metastases usually happen in isolation or together with extra-cranial metastases? Are you able to present data from the selected studies? Do you think this is a very late stage presentation or includes both early+ late?
2. Table 1. Number of GC patients, are those supposed to be commas instead of period/decimal points?
3. Figure 2. Incidence. Are those supposed to be period/decimal instead of commas? If it is %, then it should be presented as %, otherwise it should presented as table.
4. Table 2 might look better if you reverse the symptoms as on left.
5. Maybe discuss her2 drugs that have CNS penetration? There are several that are used in the breast cancer space, that is still under investigation for GE cancers.
6. Line 329, the sentence seems a bit confusing, maybe state more clearly that you mean the survival is poor with BM.
Round 2
Reviewer 1 Report
Comments and Suggestions for Authors
The authors have considerably improved the article and they have followed all the proposed recommendations.